# Exploring the experiences and views of doctors working with Artificial Intelligence in English healthcare; a qualitative study

**Shaswath Ganapathi**[1]*, **Sandhya Duggal**[1,2]

**1** University of Birmingham Medical School, Birmingham, United Kingdom, **2** The Strategy Unit, Midlands Lancashire Commissioning Support Unit, Leyland, United Kingdom

* shaswath@hotmail.co.uk

## Abstract

### Background

The National Health Service (NHS) aspires to be a world leader of Artificial Intelligence (AI) in healthcare, however, there are several barriers facing translation and implementation. A key enabler of AI within the NHS is the education and engagement of doctors, however evidence suggests that there is an overall lack of awareness of and engagement with AI.

### Research aim

This qualitative study explores the experiences and views of doctor developers working with AI within the NHS exploring; their role within medical AI discourse, their views on the implementation of AI more widely and how they consider the engagement of doctors with AI technologies may increase in the future.

### Methods

This study involved eleven semi-structured, one-to-one interviews conducted with doctors working with AI in English healthcare. Data was subjected to thematic analysis.

### Results

The findings demonstrate that there is an unstructured pathway for doctors to enter the field of AI. The doctors described the various challenges they had experienced during their career, with many arising from the differing demands of operating in a commercial and technological environment. The perceived awareness and engagement among frontline doctors was low, with two prominent barriers being the hype surrounding AI and a lack of protected time. The engagement of doctors is vital for both the development and adoption of AI.

### Conclusions

AI offers big potential within the medical field but is still in its infancy. For the NHS to leverage the benefits of AI, it must educate and empower current and future doctors. This can be achieved through; informative education within the medical undergraduate curriculum,

**Data Availability Statement:** For this qualitative work, the data is the individual full transcript of all 11 interviews. Making the data publicly available would not be appropriate as members of the public would be able to identify the participants view the transcript, and the participants signed up to the

research project with the expectation they their identity would be anonymised and could not be identified on publication. As such, restrictions on the data have been imposed by the University of Birmingham Ethics Committee and requests for data access can be made to Mark Exworthy (m. exworthy@bham.ac.uk).

**Funding:** The author(s) received no specific funding for this work.

**Competing interests:** The authors have declared that no competing interests exist.

protecting time for current doctors to develop understanding and providing flexible opportunities for NHS doctors to explore this field.

## Introduction

There is considerable excitement internationally surrounding the applications of Artificial Intelligence (AI) in healthcare, with the National Health Service (NHS) having ambitions to be world leaders [1]. Developments such as the Academic Health Science Network (AHSN) AI initiative, NHSX and the £250 million investment for the NHS AI lab are important steps in the NHS realising this ambition [2–5]. In addition, over £85 million has been awarded to 80 applicants through the 'AI in Health and Care Award' across 70 NHS sites since September 2020 [6].

AI compromises of multiple subsets [7], with *machine learning (ML)* and *deep learning (DL)* offering methods to systematically extract insight from the increasing complexity and quantity of healthcare data. The successful integration of AI tools into the delivery of healthcare can help healthcare systems in delivering the quadruple aim [8, 9]: improving experience of care; improving the health of populations; reducing the per capita cost of healthcare; and improving the experience of providing care [10, 11]. Topol outlined how the benefits would be impactful at three levels [8]:

- Clinicians—primarily assisted by rapid diagnostic aid in patient management (particularly image interpretation e.g radiology, pathology, ophthalmology), and automation of repetitive administrative tasks to free up time.

- Health systems- operational applications in backend systems which are non-patient facing that improve workflow, and a reduction in variability and in medical errors.

- Patients- process patients' health data to empower them in their own health promotion.

The four areas of focus for AI technology in healthcare as highlighted currently by responses in the 2021 NHSX survey are: diagnostics, remote monitoring, triage and population health [12]. This survey included 368 participants across healthcare in the UK, with 197 participants working as developers of AI for health and care.

However, despite increasing investment and research [13], there is surprisingly limited application of AI systems in clinical practice and recent systematic reviews have shown the majority of research studies are retrospective, lacking in transparency and at a high risk of bias [14, 15]. The slow translation and adoption of AI within the NHS and other healthcare systems can be attributed to several factors. Global issues relating to the science and safety of AI systems [16], which also include ethical and medico-legal issues are still being addressed [17, 18]. The NHS also faces further system-level challenges in technology innovation, implementation and adoption, such as fragmentation, underfunding, legacy IT infrastructure and ability in forming partnerships with the private sector [19–22].

In 2018, the AHSN conducted a national survey to identify how AI can be harnessed within the NHS. It included 106 thought leaders and AI pioneers working across the AI ecosystem in England. In addition to 'capacity and capability to deliver scope', 'clarity around ownership of data' and an 'ethical framework to build/ preserve trust and transparency', participants identified the 'education of healthcare professionals' and 'engagement of healthcare professionals' as two of the top five most important enablers of AI in the NHS; with the latter being the most important [23]. This is congruent with the recommendations from the 2019 Topol Review,

which was commissioned to explore the implications of a digital NHS on the workforce and how it would change their way of working. Topol concluded that in order to leverage the full benefits of AI, four conditions have to be met in relation to the workforce [24]:

1. Need for time and willingness to adopt new technology

2. An understanding of the technology

3. Well-designed technology meeting user need

4. Workplace support to maximise the potential of the technology.

The influence doctors and other clinicians may have in the adoption and sustaining of AI tools could also be inferred from literature regarding other technology-supported services in healthcare. Wade et al. conducted a qualitative study of 36 Australian telehealth services with the view of intentionally identifying a single point of intervention, rather than a multifactor theory, to offer practical advice that would have the maximum impact on implementation. They proposed clinicians' acceptance explains much of the variation in the uptake, expansion and sustainability of telehealth services [25]. Whilst this is not the only factor affecting the outcome, they found the clinicians that accepted telehealth would continue to supply it even if the demand was very low, there was pressures on the clinical workforce or if the technology was problematic. A sociological study conducted by Greenhalgh et al. supports this idea. A key finding from their investigation of why and when healthcare staff do not adopt nationally mandated information and communication technology (ICT) systems, was that the single most important determinant of whether a new technology-supported service succeeds or fails at a local level may be the acceptance by the workforce using it [26]. They reported that local staff, 'champions', appear key in persuading their colleagues that the technology-supported service is safe, effective, and professionally appropriate.

Given that doctors should be major stakeholders in this discourse, it is concerning that evidence suggests there are fears among doctors and other health professionals that AI may replace them and undermine trust [27]. Moreover, a web-based survey amongst 720 NHS General Practitioners (GPs) reported a disconnection between the expectations of frontline GPs compared to those of bioinformaticians and experts working with AI for healthcare [28]. Blease et al. proposed this may be due to a level of disengagement with the literature, and suggested further in-depth qualitative work to explore this.

Internationally, the awareness amongst doctors is a mixed picture. Quantitative studies have found that awareness was not widespread amongst doctors and medical students [29–32]. A 2020 survey found that American physicians' awareness of AI tools had considerably increased since 2016 [33]. A theoretical model developed by Greenhalgh et al. identified 4 elements at which clinician resistance can impede new health care technology: resistance to the policy reflected in technology, resistance to the sociomaterial constraints of the new technology, resistance to compromised professional relationships, and resistance to compromised professional practice [34]. It is unclear if this theory would also apply to AI technology, but the lack of education amongst doctors could foster negative views towards AI and therefore limit its translation.

It is important to note that there is a lack of universal consensus in what engagement of doctors with AI entails. For example, it was not clearly defined in the AHSN survey despite being identified as a key enabler of AI translation. However, since AI is largely in the development phase, rather than widespread clinical use, the authors have chosen to define it as doctors working with AI, as opposed to engaging in educational activities such as conferences or courses to further personal understanding of AI. There is limited literature exploring the

barriers for NHS doctors in engaging with AI. However, a number of barriers have been identified in the context of adoption of digital health technology (health tech) innovation, which could also pertain to AI engagement. The Nuffield Trust conducted a report to explore why the adoption of medical technologies is low in the NHS [20]. Similar to Asthana et al., they noted a number of cultural issues amongst clinicians that can act as barriers [19]. These include risk aversion, resistance to change and lack of entrepreneurial spirit. Greenhalgh et al proposed that resistance among clinicians may be due to perceived 'hidden work', where technology ultimately increases their workload [35]. These ideas link with the findings of Maguire et al, which suggested that technologies need to fit with the values, priorities and routine of staff for adoption [36]. Doctors may also be wary of the intentions of the private sector, which may discourage some from taking opportunities in industry [19].

Another key issue highlighted was that technology is largely supply-driven and top-down in the NHS, rather than the problems being identified by clinicians, and subsequently, they may see this as a threat to their professional judgement and autonomy [20]. Stakeholders from the Nuffield Trust roundtable suggested co-development with clinicians would resolve this. Collaborative designing involving doctors is also considered to be important for patient safety when developing AI tools, as their 'soft intelligence' [37] and medical expertise would provide oversight for the technology professionals who lack the clinical knowledge [23, 38, 39].

Whilst the involvement of doctors is considered to be important in the development of AI, it has been highlighted that it is unclear whose responsibility it is to innovate, and that there are a lack of incentives to encourage innovation from clinicians [20]. There is also a lack of time available for doctors to interact with these technologies and consider how they can improve healthcare [20].

Although being an emerging field, the increase in opportunities for NHS doctors to become involved with AI [40], and more broadly the digital health field, such as the Topol fellowships, NHS Digital Academy roles and the NHS clinical entrepreneurship programme has been encouraging for the development of the field [41–43]. Outside of the NHS, it is unclear what the extent of available opportunities are. There are reports of doctors pursuing academia posts and postgraduate qualifications in AI-related subjects, as well as some working within industry, for larger tech organisations such as Babylon and Deepmind, or as part of small and medium-sized enterprises (SMEs) [44, 45].

The majority of literature on doctors and industry collaboration [46–49] concerns the pharmaceutical industry or are regarding doctors' views on industry collaboration rather than the motivators, barriers and personal experiences of doctors innovating and developing health tech devices in industry. To date, there are limited qualitative studies that explore the experiences of doctors who are actively working with AI and developing this field. Laï et al conducted a qualitative study on doctors regarding AI, however, they interviewed French stakeholders on their perceptions of AI in healthcare, which included 13 doctors that were purposively selected due to a previous interest in AI rather than direct development [50].

Therefore, this study aims to address this gap by exploring the experiences of doctors working with AI in English healthcare. For the purpose of this study, 'working with AI' describes doctors who are directly involved in the development or delivery of AI, within a healthcare or industrial setting, in addition to or instead of their clinical role. The authors have identified these group of doctors as key drivers for the development, implementation and the eventual widespread adoption of AI tools within healthcare.

The research objectives of this study include an exploration of; the facilitators for doctors working within the AI field in the NHS, the challenges they have experienced, and their views on how to increase acceptance and engagement of NHS doctors nationally.

**Table 1. Recruitment framework.**

| |
|---|
| 1) Doctors working with AI in-house of the NHS in addition to clinical practice |
| 2) Doctors working with AI externally of the NHS in addition to clinical practice |
| 3) Doctors who are no-longer clinically practicing and are now working with AI full-time. For example, partners of AI health tech start-ups, full-time employees of technology companies and consultancy roles |

## Methods

The following framework (see Table 1) was used to aid recruitment. Potential participants were excluded if they had not worked with AI in English healthcare. The literature discussed above used the term clinicians and doctors interchangeably. However, it is important to note that the term clinicians compromise a range of healthcare professionals, such as nurses and physiotherapists, reflecting different roles and experiences. The authors refer to 'doctors' rather than 'clinicians' in the rest of this paper, which refers to doctors practicing healthcare on the 'frontline' and are or have been patient facing, and best described those who were included in the research.

Purposive sampling was used in a three-stepped recruitment strategy: 1) Individuals contacted either by social media (Twitter and LinkedIn) or email (n = 8); 2) Snowball sampling (n = 0); 3a) Recruitment posters shared on Twitter and LinkedIn platforms (n = 1) and 3b) Identifying gatekeepers from NHS networks and private sector companies (n = 2). Those registering interest were then provided with a participant information sheet and after collecting written e-consent, participants chose a time and date suitable to them via VoIP (Voice over Internet Protocol) technology mediated platforms. Although the onset of the COVID-19 pandemic heavily disrupted the availability of doctors, a total of 11 participants from varied backgrounds across primary care, medicine and surgery (see Table 2) were recruited.

Semi-structured individual interviews were conducted from March to April 2020. The use of a topic guide ensured a degree of consistency but also allowed the researcher to probe further on emerging themes to capture authentic insights from each individual. It included questions on the personal journeys and facilitators for the participants to working with AI, their

**Table 2. Participants key characteristics.**

| Participant | Role | Clinical-tech split | Number of years in field | AI experience |
|---|---|---|---|---|
| 1 | Foundation training | Full-time tech | 1–5 | Employee in AI health tech company |
| 2 | Consultant | 50% clinical: 50% academic/industry | 5+ | Academia and industry partnerships |
| 3 | Registrar | Full-time tech | 5+ | Founder of health tech consultancy business, academia, policy, past roles in other health tech companies |
| 4 | Foundation training | Full-time tech | 5+ | Industry partnerships and digital consultancy |
| 5 | Undergraduate medical degree | Full-time tech | 1–5 | Graduate scheme in digital consultancy in tech company |
| 6 | Consultant | Full-time clinician, working with AI in spare time | 1–5 | Founder of a start-up |
| 7 | General Practitioner | 20% clinical: 80% tech | 5+ | Current medical director of AI for a health tech company, previously working in another AI health tech company |
| 8 | Junior doctor | Full-time tech | 5+ | Policy, industry partnerships, head of an accelerator programme |
| 9 | Surgeon | Full-time tech | 1–5 | Employee in AI health tech company |
| 10 | General Practitioner | 60% clinical: 40% tech | 1–5 | Policy, ethics, industry partnerships |
| 11 | Foundation training and locum | Full-time tech | 5+ | Employee of health tech company, previously employee in another AI health tech company |

experience and challenges in this field, their views on AI in relation to the NHS and frontline doctors and the engagement of doctors in this discourse.

Audio-recordings were transcribed and analysed using Braun and Clarke's six-phase thematic analysis, with Nvivo software being used to facilitate the coding process. The coauthor randomly selected and independently coded one transcript, and codes were compared and any inconsistencies were discussed.

### Ethical issues/statement

Before recruitment began, ethical approval was sought and received from the University's Internal Research and Ethics Committee at the University of Birmingham. The two biomedical ethics principles of autonomy and non-maleficence were carefully considered in this study. In order to ensure participants were well informed they were provided with a participation information sheet outlining the role of the research, benefits, risks and withdrawal terms. Following any questions, all participants voluntarily consented to participate. Written consent was collected electronically and verbal consent was confirmed at the start of each interview. To avoid any harm or consequences to participants as a result of their views, all details are anonymous and kept strictly confidential. Data security conduct was in line with the University of Birmingham's Code of Practice for Research and General Data Protection Regulation.

## Results

Three themes were generated which will be presented in turn with data extracts to illustrate their significance: the route to working with AI; challenges of working with AI; frontline clinicians in the AI discourse.

### The route to working with AI

All participants reported having minimal mentorship and instead largely relied on their own intuition to get their opportunities, as there was, and still is, a lack of a structured pathway to enter this field. A common motivator for choosing to work with AI was the ability to have a scalable impact. In particular, five had expressed frustration with the restrictions of the healthcare system and now felt empowered in their new roles to effect change:

> "When I was in the NHS, no matter how good the people were, they were still working in that system, and I felt like a tiny cog in a huge machine incapable of actually instilling changes that I thought could make the patient and doctor experience so much better" (Participant 1)

Another facilitator for six doctors was their interest in technology from childhood, with only two having experience in AI coding. The other five doctors had only developed an interest after realising the potential of combining their clinical knowledge with AI. Three participants felt that as frontline doctors, their involvement was important to help find clinical value whilst protecting patient safety:

> "If we don't have doctors who care about patients on that side, then we will only see corporate interests expressed. So, I thought I could be a force for good in this space" (Participant 10)

Over half of the doctors had been deliberate in seeking opportunities and exposure in the AI and broader health tech field, either full-time or in addition to clinical work. In contrast,

although the other four doctors were considering alternative options, they suggested they may have never entered the field without an element of serendipity:

> "A friend of mine posted on Facebook saying he was working at a health tech company that needed more doctors. Literally I wouldn't have applied had that not happened" (Participant 11)

All participants planned to continue working within the AI health tech space but differed on the clinical-tech balance they would pursue. The four part-time doctors felt retaining their clinical practice was important to give them credibility as a clinical expert but also because their identity and strengths were that of a clinician:

> "There are some very bright people in the machine learning sphere and there is no way I can keep up with them . . . but my understanding of the clinical aspects is what makes me really useful" (Participant 2)

A reduction in stress was a shared reason for not reverting back to their full-time clinical roles. They explained that their AI roles may require work outside of their contracted hours, which could be demanding, but this was less stressful than pressures in clinical practice. One commented on enjoying his clinical sessions much more after reducing his clinical hours. All seven doctors now working full-time in industry also conveyed being more stressed in their clinical roles, with three in fixed salaried roles within tech companies reporting an improved work-life balance:

> "I now work predictable hours, I have time to go to the toilet when I want to, I have time to eat my lunch when I want to, I have time for my family and friends and I do not feel anxious at my work as well" (Participant 1)

Four participants, despite an increased workload, cited that they now experienced much greater job satisfaction largely due to greater autonomy and ability to express their affinity for innovation and entrepreneurship. Two even felt their identity had shifted from that of a clinician, to a health tech professional. Although there were certain aspects participants missed about their clinical practice, most notably the patient contact, it was not enough to motivate them to return to clinical practice:

> "There's an immediacy to medicine in terms of the rewards you get . . . patients are so thankful for everything that you do . . . there is a camaraderie around staff in the NHS that you don't really get in any other workplace . . . but those elements don't override the control I have over my own life right now" (Participant 8)

### Challenges of working with AI

The system-level barriers to developing and implementing AI that were discussed earlier [19–22] had personally impacted doctors to varying degrees due to the remit of their roles. These included: lack of a regulatory standard; access to patient data; NHS interoperability and IT infrastructure; and difficulty for tech companies (particularly SMEs) in establishing partnerships with health providers due their limited capacity to provide evidence. The consensus among all doctors was without these issues being addressed, the rate at which AI innovation and implementation happens would always be limited.

A number of additional challenges revolved around having to operate outside of a clinical environment. Initially, seven doctors experienced steep learning curves whilst acclimating to

the technological and commercial environment due to their limited skillset and experience. Nine doctors highlighted the need for greater comfort with uncertainty and bolder risk management, since unlike best practice guidelines in medicine the AI field is largely unchartered territory. Two doctors who had made the successful transition to tech full-time, cautioned that this path, particularly involvement in start-ups or as founders of companies, also brought risk in the form of less job stability compared to their NHS jobs:

> "As soon as you go outside of the NHS, the world is a lot more of a competitive place and you have to prove your worth, it is not guaranteed that you will go up a pay grade each year and all this type of stuff" (Participant 3)

In contrast to the sharing of knowledge that exists within clinical networks, three doctors commented that many in the AI field are working in silos and are reluctant to share their experiences since it is commercially valuable information. One participant described how doctors working in the health tech space are subject to different expectations and demands than what they are used to:

> "Companies will have targets to meet, contracts to win, so you have to get involved in the capitalist business world as opposed to a very socialist NHS. There's a real kind of friction there definitely for some people who try and make the jump" (Participant 8)

Collaborating within a new type of multi-disciplinary team where many have competing interests resulted in another tension between the technological mindset of 'move fast and break things' and the medical mindset of 'going slow and safe'. All participants conveyed the crux was that tech professionals lack a clinical background and so an appreciation of healthcare and the importance of collecting clinical evidence. Additionally, although doctors must raise safety concerns when necessary, they can be at fault for holding up the "red flag" [9] at every point in product development.

> "Going from an academic environment and then using real world data on live patients are two totally different scenarios . . . You can have doctors who want prospective randomized controlled trials and then you have the world of tech that don't have profit cycles that allow for it. So, there is a tension in those two cultures and I've seen technologists very frustrated that doctors aren't willing to do things quickly" (Participant 10)

> "I think what happens is doctors are very quick to say no to something because it doesn't sound right to them. But you should be able to play with an idea long enough to really assess its merits before dismissing it" (Participant 9)

All of the participants spoke of the culture clashes and spoke about how "medical voices [in certain companies] have historically felt unheard" [9]. However, the level of conflict they personally experienced varied. Seven conveyed the narrative was not a "it's us versus them" [3] and rather than being a clash, it was a difference of opinion that could be resolved. However, one participant had felt "undervalued and underrespected" [11] in the first company he worked in, and became particularly emotive when recounting his difficulties:

> "There were a lot of moments where it was an uphill battle in terms of explaining to people why clinical safety is important and why we can't just release things" (Participant 11)

Doctors offered a number of shared factors for overcoming the culture clash, which can be split into two levels. The first is at the level of the organisation, with the majority expressing the importance of companies fostering a "culture of openness and collaboration" [5], where clinical values are at the forefront. One suggested employees recruited from different domains should already have an appreciation of needing to do things differently in healthcare and three conveyed this responsibility lied upon the senior leadership team to align incentives to reward good behaviours:

> "In terms of overcoming it, it's a culture thing and it has to come from the very top down, and organisations that do it well, they'll have it placed in from the very top that we can't be like a tech company, we have to be a healthcare company" (Participant 4)

The second was at the level of the individual relationships doctors have with other professionals, which all participants stated must be built on trust. Some mentioned that demonstrating a "genuine interest in tech" [3] and appetite to learn had helped earn trust from colleagues. The majority shared the general idea of establishing a common goal, respecting different perspectives and building a "common language" [4] between the disciplines. An interesting critique raised by three participants, was doctors could be at fault for lacking humility and failing to appreciate the expertise and importance of the other disciplines within the multi-disciplinary team.

### Frontline clinicians in the ai discourse

The participants shared the view that doctors as a body of medical professionals had a responsibility in developing the future of AI, but agreed that it is ultimately an individual's decision to enter and work within the AI healthcare field. However, there was caution that we should aim for quality over quantity:

> "Yeah, of course they do [have responsibility], but not all doctors do. What we do not need is too many cooks. What we need is people who know what they are talking about driving it" (Participant 3)

One participant felt it "would be dangerous to put individual responsibility on doctors or indeed a population of doctors to change the system" [8], because this was not part of the job description when they decided to pursue a career in medicine. Instead, responsibility should fall on Health Education England to ensure they hire and recruit doctors, and other healthcare professionals with varied backgrounds, starting from undergraduate level, with the expectation of driving these changes.

Participants explained how the engagement from doctors was important for two reasons. Firstly, their experience is key in not only developing safe AI products but also identifying solutions with the highest value propositions. Two participants described how they felt the current process is flawed:

> "I think at the moment it's coming from either what people see as being profitable or what they see as being technologically feasible, but neither of those necessarily solve what are the biggest problems for the NHS" (Participant 11)

Secondly, co-development would ensure the integration of AI products within clinical workflow is considered; a vital enabler for adoption that some doctors felt was often overlooked:

"I think the easiest way to [get buy-in is] to have clinicians in-house who are helping to shape the products and then closing the loop by sort of speaking to the users of the points of deployment to find out what's working and what isn't working. That 100% needs to take place. And I've seen it happen where products are sort of built for clinicians without clinician involvement and they are not good and they're just not taken up" (Participant 7)

Several participants expressed their vision for the future, and how at a minimum doctors should be able to critically evaluate claims regarding AI products, with one proposing it could be unethical to be unaware and not use AI if it would result in the best outcome for the patient. Two participants spoke about how doctors might be at a disadvantage if they lacked a basic understanding of the impact AI and digital health.

"What you'll find is the technologies and ways of doing things change at such a fast rate you won't be able to keep up . . . things will be done to your patients that you don't agree with and you had an opportunity to be involved in the development of those systems and you didn't take it" (Participant 10)

Participants noted the increased excitement surrounding AI but felt there was too much hype, highlighting limited high-quality research and its translation to clinical impact. Five felt it was important to clarify that AI is not a panacea and instead would be used in very specific use-cases to augment doctors, with simpler digital health technologies sometimes providing the better solution. Four had observed how the concept of AI was commonly misused, sometimes even intentionally in commercial settings to gather more interest. One participant explained how this hype "runs the risk of misinformation which directly leads into awareness" [10]. The most harmful misconception being that AI tools would be a form of *General AI* would replace doctors, rather than *Narrow AI* [51], which understandably leads to anxiety and resistance:

"[doctors] thought AI was some sort of an autonomous decision making technology, but it is not. It really is data in, data out, someone has to deal with the output in a human way . . . it hasn't replaced a single doctor anywhere, because just providing an output doesn't change the fact that this needs to be put in the context of a medical care pathway" (Participant 5)

Five participants conveyed how ignorance among doctors can also form resistance as there are fears over how the analytics of data might affect their clinical autonomy and how it may change the way they practice medicine. Two participants also identified that developing explainability and interpretability of black-box algorithms would be important in gaining doctors' trust.

Some noted that whilst it was promising that opportunities within the NHS were increasing for interested doctors and other clinicians, there was still limited exposure and opportunity for the majority of the workforce. The obvious predicament many raised is that the financial and staff pressures on the NHS makes it difficult to gain funding to increase opportunities and pull clinicians out from clinical sessions. However, a couple participants felt an increase in fixed opportunities is necessary, in order to translate the potential of AI within the NHS:

"The NHS's most valuable resource is its workforce. AI's strongest potential in the NHS is as a workflow accelerator to make its workflow more effective. And the utilisation of our workforce being so high is that our workforce cannot spend time to develop additional

skills, so the NHS is relying on third party and industry to develop these things, this to me is the fundamental problem" (Participant 2)

Several highlighted how, with the exception of General Practitioners (GPs), there was a lack of flexibility for doctors to work in different industries whilst continuing their clinical training. Six participants stressed flexible career options are necessary not only to support those interested doctors in developing vital skills they can bring back to the NHS, but also to help retain NHS workforce being lost to industry:

"And then even beyond [AI] the other aspect of building or running a business and building products, there is a huge lack of knowledge amongst clinicians . . . it is going to remain the individual clinicians responsibility to learn that themselves . . . you risk losing clinicians from the frontlines because they will go off, and like myself, like some of my colleagues, drop down clinical sessions significantly, to explore these other non-clinical roles." (Participant 7)

Another solution to increase engagement was to stimulate the next generation of doctors from the undergraduate level. Participants had different views on what content should be covered within medical school, when, and whether it should be in a formal or informal capacity. An important educational distinction that transpired was between providing the awareness and basic understanding of the topic, versus, the opportunity to learn practical skills for innovation. One participant divulged that *"sometimes as clinicians we can have blinders on, to like our immediate surrounding and environment"* [9]. This idea resonated with others, who conveyed the idea of planting a seed to allow medical students to decide if they want to explore this later during their careers:

*"I think this is where you have to sow the seeds in, because by the time you become a clinician, you're too far in. You've got fixed ideas and it becomes very hard for you to then change your mindset"*- participant 6

Some participants focussed solely on AI-related content, however, others felt that the broader field of digital health technologies should be addressed and this should also be done in a formal capacity:

*"To me I think AI is almost a sort of red herring in all of this and I think it should be digital technologies that for me are what's important"*- participant 7

With the exception of two participants, they all believed it should be introduced immediately:

*"it should be now because by the time students become fully qualified doctors to consultants, this technology* [AI and machine learning] *will be in full practice"*- participant 3

## Discussion

**T**his study aimed to explore the experiences of doctors working with AI in English healthcare, focussing on: the facilitators for doctors working within the AI field in the NHS, the challenges they have experienced, and their views on how to increase acceptance and engagement of NHS doctors nationally. The facilitators and personal experiences of doctors shaping the discourse in the AI field have not been explored before, and the findings from this study addresses this gap and contributes to the growing field.

Two main themes emerged; 1) an overarching motivator for entering this space was the ability to scale their impact to patients and 2) the current pathway to begin working with AI is unstructured. The other motivators and facilitators differed. Reduced job satisfaction had pushed doctors who were now working in industry full-time to pursue alternative careers. They now enjoyed greater job satisfaction due to a mixture of factors: greater autonomy, improved work-life balance and a career better suiting their interests and skillset. Doctors who were working with AI part-time in addition to their clinical role, believed their clinical role utilised their strengths and entered the AI field due to a sense of responsibility to shape development to protect patients' interests. With the engagement of doctors identified as a key factor in the translation of AI within healthcare delivery, these findings can help to identify how to attract more doctors to work with AI. As highlighted by the different motivators and facilitators, the NHS and health systems should build a structured pathway and be cognisant when promoting opportunities that a variety of doctors from different backgrounds may be interested in pursuing a role in AI. This will help encourage a diversity of doctors to be included in shaping this discourse.

In addition to the system-level barriers to AI implementation that have been discussed in literature, a significant finding was the challenges that doctors can experience due to operating outside of their typical clinical environment. These include a lack of skillset, lack of collaboration and sharing of knowledge between health tech companies, less job stability and the uncertainty of navigating through an unchartered field. Some doctors conveyed certain skills could only be gained through experience and collaboration with professionals from other disciplines. This further supports the recommendation in the Topol review that the NHS should form partnerships with industry to provide opportunities for the NHS workforce to gain experience and skills, that will benefit the NHS [24]. This may also be important in terms of retaining NHS talent. An increased proportion of doctors are leaving the NHS due to burnout [52]. There is no data on those leaving to join industry, although this is likely to represent a small percentage of those leaving. Despite this, these doctors would represent a minority of those with specialist interdisciplinary knowledge that would be well placed to work on further development and eventual implementation and adoption of AI tools. Therefore, while providing these opportunities would remove staff from clinical roles, the NHS and related bodies must realise that providing flexible jobs for doctors to explore non-traditional clinical routes within the NHS, and work in specialist roles, could help retain the NHS talent that are being lost to full-time industry roles. Moreover, these doctors could be key 'local champions' described by Greenhalgh et al to other frontline doctors, and help build trust and acceptance towards use of AI tools [26].

The combination of working in a capitalist commercial setting and working alongside non-healthcare professionals who are used to a 'move fast and break things' mantra can bring tension. In particular, clinical values should be at the forefront for patient safety and identifying the biggest value propositions, supporting the view expressed in reports and articles [23, 38, 39, 53, 54]. The findings revealed the building trust with non-healthcare professionals was a solution to manage these culture clashes. This can be fostered in a number of ways: recognising and respecting the different expertise, developing a common language, establishing shared goals and demonstrating an interest to learn. Going forward, healthcare organisations encouraging the development of doctors working with AI should make them aware of this culture clash and share insight from doctors already working in this field to support their transition. Industry organisations should empower doctors working within their setup to raise safety issues and take steps to address a culture clash.

It is important to note the significant hype and misinformation surrounding AI, and in fact, progress is still in its infancy, with limited robust evidence conducted in real clinical

practice and limited translation in healthcare. This is concordant with the views of doctors interviewed by Laï et al. [50], and resonates with the sentiment in existing literature [16, 22]. It also supports the reports from NHS-related organisations [55] that have highlighted that AI should not be heralded as a silver bullet.

The findings presented here suggest that frontline NHS doctors need to be significantly engaged with AI for its development, adoption and integration, which supports previous recommendations [24, 55–57]. It is therefore problematic that all participants perceived low awareness and understanding among frontline doctors, as suggested by Blease et al and anecdotal evidence [29–32]. In fact, the findings reveal disengagement can be categorised to three groups: 1) those resistant and sceptical; 2) those unaware or indifferent; and 3) those who may be interested in working in the AI healthcare field but lack time, mentorship and flexible opportunities to enter this field. The first group largely stems from the misconceptions propagated by the hype [27]. These included worries about AI replacing doctors, the risks of AI, resistance to change, and fear over losing their clinical autonomy, mistrust in partnering with industry; all having parallels to previous findings on barriers to adoption of technologies in healthcare [19, 20, 35, 36]. While some in the second group may prefer to focus on clinical work, it is also possible that they are simply unaware. Industry and healthcare organisation should avoid propagating hype and provide more accurate depiction of AI use-cases to promote translation of AI.

A limitation of this study was the small sample size. A total of 24 doctors had either expressed interest or had an interview arranged, but could not participate due to limited time from responding to the COVID-19 pandemic. A number of doctors from particular backgrounds could not be interviewed, for example, the majority of this sample compromised of doctors who were working with AI in a full-time capacity. This sample also did not contain any doctors working with AI in-house of the NHS, which was a group within the recruitment framework (Table 1). Female underrepresentation in tech has been highlighted as an issue [58] and although the researcher intended to have similar ratio of male to female participants, only three females were included in this sample. Due to there not being an established doctor developer role, there is likely a range of roles and experience that could yield important data to contribute to this evidence base. Although there was diversity within the sample, these findings may not be generalisable to the experiences and views of all doctors working with AI.

There are three areas of research that are warranted to better understand how to increase awareness amongst frontline clinicians, how to encourage more doctors to work with AI and how to best support them: 1) As recommended by Blease et al, further qualitative research should be conducted interviewing doctors who are not directly working with AI, to explore their attitudes and perceptions on AI; 2) qualitative research on the views and experiences of other non-clinical professionals developing AI tools with doctors within these interdisciplinary team; 3) qualitative research on the views and experience of the teams managing the implementation of AI tools and the healthcare professionals adopting them within their clinical workflow.

The key recommendations in Table 3 revolve around better education for both current and future doctors, providing more opportunities and fixed time for those interested and, a call to industry to empower doctors within their setup. The findings from this study demonstrate that engagement with AI is still in its infancy and whilst addressing wider translational challenges, the NHS and related bodies must ensure they dispel misconceptions and build a receptive context for adoption by engaging current and future workforce, whilst also providing opportunities for interested doctors to develop new skillsets and shape this discourse.

**Table 3. Recommendations.**

| NHS and Medical Bodies | • Address the negative narrative surrounding AI among doctors to build a receptive context primed for the NHS to benefit from AI.<br>• State the role and expectations of frontline doctors in the AI discourse.<br>• Produce educational material to increase necessary understanding and skills among doctors for them to evaluate and use AI, with protected time to meaningfully engage with these resources.<br>• Provide flexible careers and pathways to support doctors to gain experience and skills that can progress AI in the NHS. This may also help retain doctors leaving for alternative careers.<br>• Partner with industry to provide opportunities for doctors, that benefit both parties. |
|---|---|
| GMC and HEE | • Establish guidelines regarding the formal content related to AI and digital health that all medical schools must fulfil in the undergraduate curriculum.<br>• Provide additional opportunities for students to explore interests in AI and digital health. These may be in the form of internships, electives and intercalations. This will help develop a cohort of medical students that will fulfil specialist roles in AI and digital health in the NHS. |
| Industry and Technology companies | • Senior leadership must foster a culture of transparency and openness in order to gain the buy-in of NHS doctors and empower doctors working within their setup. This will increase adoption of AI products and allow safe integration into clinical workflow.<br>• Provide opportunities for doctors to co-develop AI solutions that have real value to the NHS. |

## Acknowledgments

The authors thank Professor Mark Exworthy for his support, as well as two anonymous peer reviewers, and journal editors for their valuable comments.

## Author Contributions

**Conceptualization:** Shaswath Ganapathi, Sandhya Duggal.

**Data curation:** Shaswath Ganapathi.

**Formal analysis:** Shaswath Ganapathi.

**Investigation:** Shaswath Ganapathi.

**Methodology:** Shaswath Ganapathi, Sandhya Duggal.

**Project administration:** Shaswath Ganapathi.

**Resources:** Shaswath Ganapathi, Sandhya Duggal.

**Software:** Shaswath Ganapathi.

**Supervision:** Sandhya Duggal.

**Validation:** Shaswath Ganapathi.

**Visualization:** Shaswath Ganapathi.

**Writing – original draft:** Shaswath Ganapathi.

**Writing – review & editing:** Sandhya Duggal.

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
