## [Decision Letter · Decision Letter 0]

2 Mar 2022

PONE-D-22-02095Exploring the experiences and views of doctors working with Artificial Intelligence in English healthcare; a qualitative studyPLOS ONE

Dear Dr. Ganapathi,

Thank you for submitting your manuscript to PLOS ONE. After careful consideration, we feel that it has merit but does not fully meet PLOS ONE’s publication criteria as it currently stands. Therefore, we invite you to submit a revised version of the manuscript that addresses the points raised during the review process.

We look forward to receiving your revised manuscript.

Kind regards,

Angeliki Kerasidou

Academic Editor

PLOS ONE

Journal Requirements:

Reviewers' comments:

Reviewer's Responses to Questions

**Comments to the Author**

1. Is the manuscript technically sound, and do the data support the conclusions?

Reviewer #1: Yes

Reviewer #2: Partly

2. Has the statistical analysis been performed appropriately and rigorously? 

Reviewer #1: N/A

Reviewer #2: N/A

3. Have the authors made all data underlying the findings in their manuscript fully available?

Reviewer #1: No

Reviewer #2: Yes

4. Is the manuscript presented in an intelligible fashion and written in standard English?

Reviewer #1: Yes

Reviewer #2: Yes

5. Review Comments to the Author

Reviewer #1: When I began reading this paper I thought it was going to be about healthcare professionals’ views of using /implementing AI in clinical practice, it is more than this. Its interesting focus is on how a particular group of people, all trained in medicine many now working in Industry ( a small group still working sometime in medicine), think about their role in the development of this technology and its potential for implementation more widely. Indeed, theme 1 of the analysis talks about how interviewees came to be working in the field of AI and theme 2 explains how different and what the challenges are to them personally working in/with industry, and some of the problems of interdisciplinary working, themes 3 was a little bit more of what I expected the paper to be about from the introductory review although there is still a focus on getting HCPS to become involved in implementation and design not just using AI . While I realise from line 115ff this is stated (although it only became clear to me once I started reading the analysis) I think you need to make it much clearer your project was looking at experiences of AI clinician developers from the outset.

That said this is interesting data about collaboration with industry in the medico-industrial complex and the paper adds to the debate about the implementation of AI in healthcare, given that it involves those who have extensive medical and knowledge of AI, they are involved in its design to a greater or lesser extent in every case. However, I think that in addition to making the purpose of the paper less ambiguous up front, it also needs a background that considers the barriers/drivers for those who work in medicine and industry, it seems to me that this in many ways is the bedrock of the paper. In relation to this the discussion really just reiterates the findings, it does not really discuss the very interesting data in light of the literature to any degree. I would have liked to see more literature cited regarding clinical industrial collaboration and moving across this boundary. I do not know this literature, but I am sure it exists.

I have some minor comments/observations

Intro: Maybe explain what Topol review is to the uninitiated.

Re ethics : I think it might be advisable to roughen the data in table one a little clearly there may be only one or two ear nose and throat surgeons who are working in the tech sector in the UK, so maybe just call them surgeon, unless the speciality is really important for subsequent points you are making (I don’t think it is) , same for other types of physicians

Discussion : Is surely the wrong way around you discuss findings of theme 3 first, but the bulk of the paper is about the ways in which these individuals moved into industry/industrial collaborations and how they experience this. As written the discussion of this aspect of the data comes after the theme three discussion, i.e how they perceive clinical colleagues will perceive and cope with this technology and what sort of tech training they need.

Does your study have any shortcomings?

some typos

Line 59 word missing; studies?

Line 86 sentence starting whilst may be missing words

Line 234 – system level barriers to what, implementation, working with AI – please expand

362 I assume you mean the concept of “AI” was used ……

Reviewer #2: I find that this paper has the potential to make a valuable contribution to the field. The authors argue that the uptake of AI in healthcare is dependent on adequate clinician engagement. They use interview data to show that there is currently an unstructured pathway for doctors to enter the field of AI and they identify the barriers preventing doctors to engage with the development of AI in healthcare. The authors recommend that current and future doctors must have the opportunity to be educated and would benefit from increased flexibility allowing for part-time industry work, without losing clinicians to full-time non-clinical jobs.

The empirics section is clear and well structured. However, I found that the introduction/literature review section would benefit from a revision. It could be a lot more focused, narrowed down, with clear definitions of terms and a clear outline and justification of the objectives of the research.

First, the introduction does not lay out clearly and specifically what the empirics section is trying to answer. The research aim is vague, and the introduction does not clarify the aims adequately. Several expressions are being used to refer to clinicians interacting with AI in some capacity:

• Engagement with AI (p.1, line 18 and 20)

• Doctors working with AI within the NHS (p.2, line 24)

• How engagement with AI may increase (p.2, line 26)

All these expressions are very vague and could refer to many things. When the authors talk about engagement or working with AI, what exactly does it refer to? does it refer to the use of AI tools in clinicians’ own clinical practice? Engagement with data science in medical education? This is not made clear in the introduction, and it is only when reaching the empirics section that the reader understands that the authors are talking about clinicians getting involved in the industry, either instead of their clinical role, or in addition to it.

Secondly, and related to the first point, instead of explaining at length in the third paragraph how COVID-19 impacted AI development and adoption, the paper could focus more on the literature addressing why AI adoption in healthcare is slow and limited, and single out and explain why the lack of involvement of doctors in the industry is a key determinant. Ultimately, there is a lack of a justification for why the authors decide to interview this group of people specifically. For example, why not clinicians working with AI in their everyday practice? If the argument is that NHS staff should be involved with AI in order to, in turn, persuade colleagues, clinicians who use AI tools are arguably also well placed to persuade their colleagues that technology-supported service is safe.

Smaller comments:

• Page 5 line 102: it is not immediately obvious how the engagement and education of healthcare professionals supports Greenhalgh et al’s theory (there is a typo in her name, no "u"). It might be helpful to add an explanatory sentence.

• Page 13 line 210: the paragraph about work-life balance does not seem directly relevant to the title of the sub-section “the route to working with AI”. This should perhaps become a different/additional section.

6. PLOS authors have the option to publish the peer review history of their article (what does this mean?). If published, this will include your full peer review and any attached files.

Reviewer #1: No

Reviewer #2: **Yes: **Aurelia Sauerbrei

---

## [Author Response · Author response to Decision Letter 0]

6 Jan 2023

We are extremely grateful to the reviewers feeling this paper does have merit following major revisions. With the help of these comments, we feel these revisions have improved the focus of this paper, stated the aims of the paper clearer and why ‘doctors working with AI’ are key proponents in this discourse, and therefore warrant this qualitative study.

We have attached a document, in addition to a cover letter, manuscript, manuscript with track changes, called 'response to reviewers'. Within this document we have detailed our response to the reviewers comments

We have also attached a cover letter with the details of the ethics body imposing the ethical restrictions and non-author contact.

We have also updated the data availability statement as recently requested on 05.02.23 and can confirm the proposed data availability statement: "For this qualitative work, the data is the individual full transcript of all 11 interviews. This data cannot be shared publicly as members of the public would be able to identify the participants via the transcripts. This would not be ethical as the participants signed up to the research project with the expectation that their identity would be anonymised and could not be identified on publication. As such, restrictions on the data have been imposed by the University of Birmingham Ethics Committee and requests for data access can be made to Mark Exworthy (m.exworthy@bham.ac.uk) for researchers who meet the criteria for access to confidential data."

---

## [Decision Letter · Decision Letter 1]

15 Feb 2023

Exploring the experiences and views of doctors working with Artificial Intelligence in English healthcare; a qualitative study

PONE-D-22-02095R1

Dear Dr. Ganapathi,

We’re pleased to inform you that your manuscript has been judged scientifically suitable for publication and will be formally accepted for publication once it meets all outstanding technical requirements.

Kind regards,

Dorothy Lall

Academic Editor

PLOS ONE

Additional Editor Comments (optional):

Reviewers' comments:

Reviewer's Responses to Questions

**Comments to the Author**

1. If the authors have adequately addressed your comments raised in a previous round of review and you feel that this manuscript is now acceptable for publication, you may indicate that here to bypass the “Comments to the Author” section, enter your conflict of interest statement in the “Confidential to Editor” section, and submit your "Accept" recommendation.

Reviewer #1: All comments have been addressed

Reviewer #2: All comments have been addressed

2. Is the manuscript technically sound, and do the data support the conclusions?

Reviewer #1: Yes

Reviewer #2: Yes

3. Has the statistical analysis been performed appropriately and rigorously? 

Reviewer #1: N/A

Reviewer #2: N/A

4. Have the authors made all data underlying the findings in their manuscript fully available?

Reviewer #1: Yes

Reviewer #2: Yes

5. Is the manuscript presented in an intelligible fashion and written in standard English?

Reviewer #1: Yes

Reviewer #2: Yes

6. Review Comments to the Author

Reviewer #1: this is much improved and a really important addition to the literature I enjoyed reading it once again.

Reviewer #2: Thank you very much for the opportunity to comment on the revised version of this paper. The authors have made a substantial effort to improve the manuscript and I am happy with the way all comments have been addressed.

7. PLOS authors have the option to publish the peer review history of their article (what does this mean?). If published, this will include your full peer review and any attached files.

Reviewer #1: No

Reviewer #2: No

---

## [Editor Report · Acceptance letter]

21 Feb 2023

PONE-D-22-02095R1 

Exploring the experiences and views of doctors working with Artificial Intelligence in English healthcare; a qualitative study 

Dear Dr. Ganapathi:

I'm pleased to inform you that your manuscript has been deemed suitable for publication in PLOS ONE. Congratulations! Your manuscript is now with our production department. 

Kind regards, 

on behalf of

Dr. Dorothy Lall 

Academic Editor

PLOS ONE